# Probabilistic Framework Allocation on Underwater Vehicular Systems Using Hydrophone Sensor Networks

**Pravin R. Kshirsagar** [1], **Hariprasath Manoharan** [2], **S. Shitharth** [3], **Abdulrhman M. Alshareef** [4], **Dilbag Singh** [5] **and Heung-No Lee** [5,*]

1 Department of Artificial Intelligence, G.H Raisoni College of Engineering, Nagpur 414008, India; pravinrk88@yahoo.com
2 Department of Electronics and Communication Engineering, Panimalar Institute of Technology, Poonamallee, Chennai 600123, India; hari13prasath@gmail.com
3 Department of Computer Science & Engineering, Kebri Dehar University, Kebri Dahar P.O. Box 250, Ethiopia; shitharth.it@gmail.com
4 Department of Information Systems, Faculty of Computing and Information Technology, King Abdulaziz University, Jeddah 21589, Saudi Arabia; amralshareef@kau.edu.sa
5 School of Electrical Engineering and Computer Science, Gwangju Institute of Science and Technology, Gwangju 61005, Korea; dilbagsingh@gist.ac.kr
* Correspondence: heungno@gist.ac.kr

**Abstract:** This article emphasis the importance of constructing an underwater vehicle monitoring system to solve various issues that are related to deep sea explorations. For solving the issues, conventional methods are not implemented, whereas a new underwater vehicle is introduced which acts as a sensing device and monitors the ambient noise in the system. However, the fundamentals of creating underwater vehicles have been considered from conventional systems and the new formulations are generated. This innovative sensing device will function based on the energy produced by the solar cells which will operate for a short period of time under the water where low parametric units are installed. In addition, the energy consumed for operating a particular unit is much lesser and this results in achieving high reliability using a probabilistic path finding algorithm. Further, two different application segments have been solved using the proposed formulations including the depth of monitoring the ocean. To validate the efficiency of the proposed method, comparisons have been made with existing methods in terms of navigation output units, rate of decomposition for solar cells, reliability rate, and directivity where the proposed method proves to be more efficient for an average percentile of 64%.

**Keywords:** underground vehicles; hydrophone; sensing unit; reliability

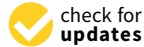



## 1. Introduction

Electric vehicles are becoming increasingly necessary in today's generation systems, with the process of E-vehicles being used not only in automobile manufacturing companies, but also in the design of underwater vehicular technology. Floating photovoltaic in the presence of PV panels is the process of studying underwater systems with solar panels. The process of improving hydrophone sensor networks is divided into three phases: hardware, controller, and operational stages. Underwater vehicles are implemented in the hardware section with sensor inputs that are handled without the use of an external controller. A loading procedure employing human and vehicle systems is combined in a single operational platform in the next phase, which is connected to control and state operation. The array of sensor subsystems will absorb the designing process and construction opportunities with respect to task requirements in both of the previously described segments. Even with all localised platform underwater vehicles, in addition to sensing capabilities, they can execute a variety of duties, such as clearing a wreck deep below the sea. However, this type

of autonomous process necessitates a large number of sensors, which can be avoided by building hydrophone systems with commercial electronic components. The main rationale for avoiding multiple units is that redundant parameters are found to be lower, ensuring a continuous development process.

The modification of modern hydrophone systems will address a number of existing system flaws, such as an insufficient decision-making mechanism. Additional duties with operational units will include data collecting and the installation of monitoring cables to check for the presence of multiple trawls. There are many procedures and utensils that are available for tracking the underwater vehicle, whereas the proposed method builds a navigational communication system that discovers the depth of the ocean using assimilated navigation technologies. If the above-mentioned inventive awareness is introduced in the field of ocean monitoring systems, then it will save many individual lives as the people travelling in the ocean can find the depth of immersion and a warning signal will be issued in case more depth is found. In addition, the underwater vehicle which travels inside the water will discover the harmful sea faunas that are present in any nearby locations. The above-mentioned navigation and discovery process will happen only if proper output points are located, and this step is termed as critical stage monitoring systems. The critical objective case will be operated at low energy as solar panels are installed with vehicular systems, and the battery operated will be designed as the water-resistant model. Moreover, the ideas will give better reliable solutions as multiple output points are tracked and connected as an internal graph where the underwater vehicle automatically travels with the marked points. As a result, hydrophone systems in this category are capable of making probabilistic judgements in any watery marks, hence meeting high objective requirements. The process of hydrophone sensor systems is depicted in Figure 1.

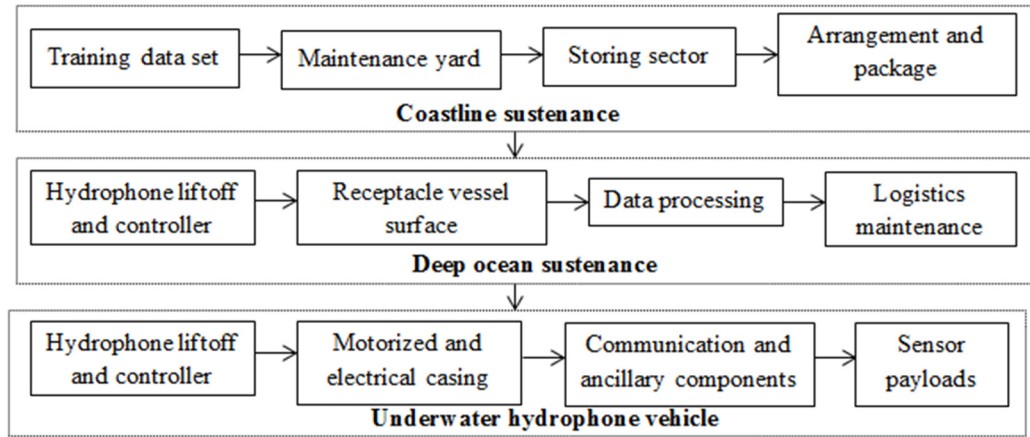

**Figure 1.** Integration of hydrophone sensors in underwater sustenance.

## 1.1. Existing Approaches

A deep examination of multiple existing methodologies has been compared with the suggested way to better understand the design aspect of hydrophones employing solar systems. In [1], the authors have examined an experimental study where the charge for operating underwater vehicles will go down as deep oceanic operations are present. Therefore, a design attempt is made using mobile aqua charging where the wave energy will be converted, thus making the procedure suitable for an autonomous surface. However, with charging points the electrical parameters are not defined; therefore, no information about the threshold levels can be obtained. Following a review of several literatures in the construction field, it is clear that one or two researchers have implemented in real time, while other cases are being investigated. The authors of [2] describe the development of a low-energy solar intellectual gadget for all terrestrial situations that uses a cloud storage technology known as the Internet of Things (IoT). When an underground monitoring system is built, however, wireless data sharing is not allowed, and the nodes that communicate

with sensing devices will move in relation to the critical point focus. Because estimating the depth of saltwater is challenging, the underwater solar boat must be designed using floating components. As a result, a floating solar panel [3] has been proposed as a cost-effective alternative to renewable energy sources in situations where the energy load demand limitation criterion is met. Despite the fact that the movement of panels is precise due to temperature and wind speed monitoring, the approach has no explanation for the data monitoring system, which is a major goal for subterranean water monitoring systems. As a result, solar panels can only float in the upper section of the ocean, while the lower part of the ocean remains in the same condition.

An autonomous subterranean vehicle [4] has been built to display the situation of underground water and monitor the condition deep inside the ocean for lengthy periods of time. The behavior-based approach, in which a neural network analyses the navigation conditions, is used to plan the autonomous vehicle. In addition, solar panels are incorporated to save more energy. However, such a created method can only measure the state of water, and the monitored data cannot be transferred to the central station in a timely manner. The data are then processed using an image processing technique [5] with the same construction design in the next stage. Additionally, colour identification algorithms for various trawls and obstacles have been integrated with a monitoring system that offers clear information for appropriate signal control. In contrast, a separate power supply for processing images is to be attached for charging it; therefore, a system has been devised for charging the sensing devices [6] using solar panels, where more energy may be saved without any expulsion. Even with different types of water technology, the same technique will perform better by monitoring characteristics like temperature, pollutants, and water volume.

Further real-time research has been carried out on Nasser Lake [7] by placing floating solar panels, where the energy of solar panels is created in an automated manner while saving a greater amount of water. The aforementioned impact saves 16 percent energy and so provides an excellent solution for underwater monitoring using floating technology. Despite the fact that energy savings are maximised, the data processing strategy is not slanted in any portion of the procedures, and such designs will only satisfy industrial needs. In response to the aforementioned worry, a practical solution has been developed to meet the demands of all persons in aquatic farms [8]. The aforementioned method focuses on multi-modal fusion technology by combining sensors into fuzzy systems with Monte-Carlo simulations used for the estimate. Because the position of the underwater vehicle is properly fixed, a navigation analysis has been performed, resulting in a suitable application solution for underwater monitoring systems. However, the method proposed for utilising many sensors is subject to a number of restrictions, making the procedure difficult in any context. A recurrent neural network has been integrated to address the complexity by estimating the state of error in the presence of filters [9]. The main goal of employing the filters is to lower the amount of noise in underground tube-shaped signals so that clear signal localization may be determined.

A prototype system has been developed to evaluate underwater systems using the perception of a gyroscopic energy scavenging device [10]. This type of small-scale technology has a high rate per minute elevation capacity while also ensuring unconstrained autonomous vehicle movement thanks to the vertical positioning axis. As a result, the model only produces useful results when it is positioned vertically, and ignores the influence when it is positioned horizontally. As a result, a hybrid system has been built by combining solar systems with robotic technology [11], and it is difficult to examine the effect present inside it owing to the lack of source stations. Fundamental results, on the other hand, demonstrate the efficacy of robotic technology when used as an alternative to traditional methods. To address the lack of charging stations, a wireless sensor technology with 10 different sensor nodes has been proposed [12]. At this time, it is assumed that wireless technologies will have a greater impact on signals with less noise consumption, and that monitoring systems will be able to reach a depth of roughly 30 metres. Then, utilising the same wireless stations, advancements in robotic technology were produced [13], and

the behaviour of cloud technology for watching such transitions with industry connections resulted in reduced resource allocation. The next step was to develop a self-triggered model using the predictive control technique [14]. The predictive control approach was created to provide closed loop system solutions using time reduction tactics. In [15], comparisons with traditional approaches were made, and the goal of adding underwater autonomous systems was discussed. All of the approaches indicated in the comprehensive assessment are unable to give information on the handling of nuclear waste found in underwater systems. As a result, a sensor-based nuclear waste detection system that ensembles for all environmental circumstances has been built.

### 1.2. Background

In the previous research work of authors [16], an intelligent decision-making approach using a machine learning algorithm has been introduced for monitoring the aquaculture fishes. The process of monitoring involves an automatic monitoring system, and, in this procedure, the underwater vehicle is not integrated. However, in the proposed method an underwater vehicle has been introduced with hydrophone sensors where the presence of aqua faunas can be monitored, and images are directed to the desired servers. The major difference between the proposed method and the basic approach in [16–25] is the way that sensors are used in the introduced systems. In the proposed technique, a probability-based approach is used with an underwater vehicle for monitoring all the necessary parameters, whereas in the basic approach decision-making algorithms are used for determining the threshold levels and the same system is rationalized for effective computations. Table 1 shows the comparison of developments with existing systems.

**Table 1.** Key methodology in conventional systems.

| Reference | Technology Developed | Purpose of Development |
|:---:|:---:|:---:|
| [11] | Wireless solar boosting | Navigation monitoring |
| [12] | Robotic technology | Maximizing the depth of monitoring systems |
| [13] | Robotic technology with wireless stations | Minimizing the resource allocation |
| [14] | Predictive control | Control of closed loop systems |
| [16] | Decision-making approach | Monitoring the physical condition of aqua faunas |
| [19] | Microelectromechanical systems (MEMS) | Minimizing the sensitivity of underwater acoustic waves |
| [25] | Election-based routing algorithm | Maximizing the stability of network |
| Proposed | Hydrophone wireless sensors | Navigation detection and deep ocean monitoring |

### 1.3. Research Gap

It was discovered that the study articles published before focused on underwater systems using various methodologies [17–37]. Despite the fact that most systems are well-established in real-time implementation, conventional systems have a minor flaw. The primary significance is that the majority of academics have not focused on vessel load, and just a few mathematical models have been stated. Subsequently, in underwater systems, most of the assets, such as water quality and fish species, will be monitored, but detection will not be possible using a squeaky tube. Furthermore, if the Global Positioning System (GPS) fails, trawlers will be unable to employ a proper navigation system in areas where the water depth is higher at centre locations [38–43].

### 1.4. Proposed Methodology

In the current circumstances, the underwater vehicular system must be converted and designed as a smart monitoring structure where all autonomous operations must be assured. Therefore, the proposed approach's intentions are to create an underwater vehicle that follows the output points which are defined in the pre-defined process; thus, only the impact points in the ocean are monitored and informed to control center. This methodology

saves more time as low impact points are not marked, thus making a suitable conversion in the location systems. Further, if the underwater vehicle is mislaid, then all output points will be gathered and at the last marked output point it will be existent. In addition, this type of underwater vehicle will provide a better trade-off between output points and the consumed energy gap where a round of 360 degrees will be controlled using multiple sensors.

The hydrophone systems are introduced through a small tube with a thicker wall than traditional approaches. In addition, the cost of such a congealed tube is lower, and the tube's length is also longer. If GPS signals are lost, trawlers can use hydrophone tubes to identify the systems, which have the added benefit of being able to be connected under the sailing boat itself. Furthermore, the planned technology will monitor the ocean's depth and allow trawlers to sail in a safe manner. New mathematical formulations with the integration of algorithmic determinations have been framed to establish this paradigm.

### 1.5. Objectives

The proposed method on hydrophones, which are implemented under the sailing boat using a tiny pipeline system, focuses on the following objectives:

- To monitor the depth of ocean and navigate the users to find protective positions;
- To find proper output points of installation at varying time periods;
- To minimize the energy consumption by installing solar panels;
- To reduce the amount of sensitivity using selective point methods.

### 1.6. Paper Organisation

The remainder of the work is organised as follows: Section 2 contains the relevant mathematical concepts for implementation. In Section 3, step-by-step integrations of proposed formulations with algorithmic decisions are discussed. In Section 4, real-time implementation is examined, and graphs are simulated using MATLAB to demonstrate effectiveness, and Section 5 closes the manuscript's perception.

## 2. Methodical Interpretations for Hydrophones

For the combined method of navigation detection (ND) and operation of deep ocean (DO), it is necessary to operate hydrophones as a new sensor unit, and the design process of NDDO will be activated by installing it under the liner. Therefore, the sensor unit must be positioned at the proper point [3] where the output points are measured. The novelty of the designed mathematical model is in the calculation of output points without any location of GPS which is defined using Equation (1).

$$\omega_i = \sum_{i=1}^{n} t_{in} + \delta_{in} + N_{in} \tag{1}$$

where $t_{in}$ denotes the exact output position of $n$ different vessels;

$\delta_{in}$ represents the bias point of varying time periods;
$N_{in}$ indicates the presence of noise in the measurement system.

Equation (1) is deliberated for reducing the amount of disregard that is present during the installation process. In addition to sensing units, a power saving solar expedient is also installed at the top of the boat; therefore, it is necessary to analyse the varying output points of solar units [23] and it is represented using Equation (2).

$$\Delta_i^{PV} = \frac{1}{\rho} \sum_{i=1}^{n} \frac{\alpha_{in}^{AC}}{\alpha_{in}^{DC}} \tag{2}$$

Here, $\frac{\alpha_{in}^{AC}}{\alpha_{in}^{DC}}$ denotes the factor that provides the de-rating percentage of DC and AC units; $\rho$ represents the energy rate that is spent for the operation of hydrophones in kWh.

Equation (2) is framed in such a way for forming the PV array system that represents a sub point co-ordinate system by varying AC and DC units. Further, after combing two output units that are represented in Equations (1) and (2), the temperature monitoring system should be implemented with the help of the hydrophone temperature sensing system [12] and it can be represented using Equation (3) as,

$$T_i = \sum_{i=1}^{n} 1 - \vartheta(S_{in} - I_{in}) \tag{3}$$

where $S_{in}$ and $I_{in}$ denote the actual sea temperature and irradiation temperature of sensor units in hydrophones.

It is required that for hydrophones using solar cells, battery capacity should be determined for significant autonomous activity [2] which is expressed using Equation (4) as follows,

$$C_{in} = \sum_{i=1}^{n} \frac{A_{in} \times LE_i}{d_i \times \tau_n} \tag{4}$$

where $A_{in}$ and $LE_i$ denote the autonomous activity of the hydroponics unit with corresponding established load at input ends.

$d_i$ and $\tau_n$ represent the decomposition battery rate that is supplied at the input unit with calculated values at the output unit.

In the attitude of hydrophones, a transducer is present where all acoustic waves are monitored with respect to changes in ranges [8], and it is denoted using Equation (5).

$$\theta_i = 10 \log \sum_{i=1}^{n} \frac{Ref_i}{Inc_i} \tag{5}$$

where $Ref_i$ and $Inc_i$ denote the reflected and incident waves at a distance of 1 m from the selected target.

If both the waves are not premeditated from the hydrophone in a designed manner, then the sensitivity parameter will be determined [12] based on Equation (6) as follows,

$$\mu_i = \sum_{i=1}^{n} \sqrt{\frac{(V_T I_{yi} r_{yi}) * (V_T I_{xi} r_{xi})}{(V_{Tin} I_{yx} r_{yx})}} \tag{6}$$

where $V_T$ represents the voltage level that is supplied at the transducer.

$I_{xi}$, $r_{xi}$, $I_{yi}$ and $r_{yi}$ denote the supplied current and inverted response of the transducer at output units for all corresponding axes.

In Equation (6), all output units are represented in a separate way, but at the data analysis stage the combined three axis representation should be represented in terms of directivity, and it can be formulated [5] using Equation (7).

$$\varphi_i = 10 \log \frac{12.56}{\int_{i=1}^{n} d(\varnothing_i) d\varnothing} \tag{7}$$

where $d(\varnothing_i)$ represents the directivity parameter at the incorporated angle of incidence.

Further, the integrated objective function (Equation (8)) is designed with both the minimum and maximum objective case where no conventional techniques have integrated it.

$$O_i = min \sum_{i=1}^{n} \mu_i cost_i \theta_i + max \sum_{i=1}^{n} C_i \varphi_i \tag{8}$$

Both minimization and maximization objectives are incorporated in the round loop system conditions where even glacial regions can also be covered. Equations (1)–(7) are framed as a new mathematical model for the design of hydrophones with solar systems to monitor the underground applications where the proposed formulations are based on

probabilistic parametric values, which are combined using a probabilistic algorithm in subsequent sections.

## 3. Integration of Prospect Using Algorithmic Determinations

The most typical method of tackling design challenges using probabilistic techniques has been discussed in this section. The main benefit of using a probabilistic algorithm is that it can solve dynamic data at both the operational and planning stages, which solves all non-optimisation problems. Furthermore, all dependability tests may be readily addressed using generic scenario case studies, and any changes in values can be simply observed if the location changes. The proposed method's probabilistic algorithm is built in the circumstance where the following equation must be satisfied.

$$\Delta_i \neq \aleph \forall P_i \tag{9}$$

where $\aleph$ represents the bounded error polynomial vector.

All bounded errors should be smaller than the acceptable polynomial time when all discrete issues may be addressed, resulting in stochastic probability scenarios, according to Equation (9). The cost of the implementation process will be minimised if the proposed algorithm is integrated, and this minimisation problem can be formulated using Equation (10).

$$cost_i = \sum_{i=1}^{n-1} E(J(cost_i)) \tag{10}$$

where

$E(J(cost_i))$ denotes the expectation values of cost function.

All the expectation values in Equation (10) are dependent on future states where they are subject to the following constraint.

$$J(cost_i) = \begin{cases} 0 \ \textit{for global solutions} \\ -1 \ \textit{elsewhere} \end{cases} \tag{11}$$

Equation (11) states that global solutions can only be found in polynomial time if the random process has no bounded faults. If any bounded mistakes are found, the process will not converge at low iteration numbers, and a value of $-1$ will be represented. A non-matched degree will be employed with biased power for different frequency ranges of hydrophones, and so the link between dependability and discriminatory can be determined as follows:

$$f_i = \sum_{i=1}^{n} \frac{1 - Z(i)}{1 - X(i)} \tag{12}$$

where

$Z(i)$ and $X(i)$ denote the error rate in terms of reliability for hydrophones using different data sets.

A hydrophone, on the other hand, should be constructed in such a way that the navigation system is clear, allowing for the installation of a greater number of nodes between the source and the destination. The equation represents the dimensional search space for the number of nodes (12).

$$\hat{N}_i = \sum_{i=1}^{n} \frac{d(s) + d(c)}{Total \ number \ of \ nodes} \tag{13}$$

where

$d(s)$ and $d(c)$ represent the distance of source to current node.

The node separable points can be identified using Equation (12) in such a way that the destination will differ until a specific distance is reached. Figure 2 depicts a flow chart of a probabilistic path finding method with interconnected formulations.

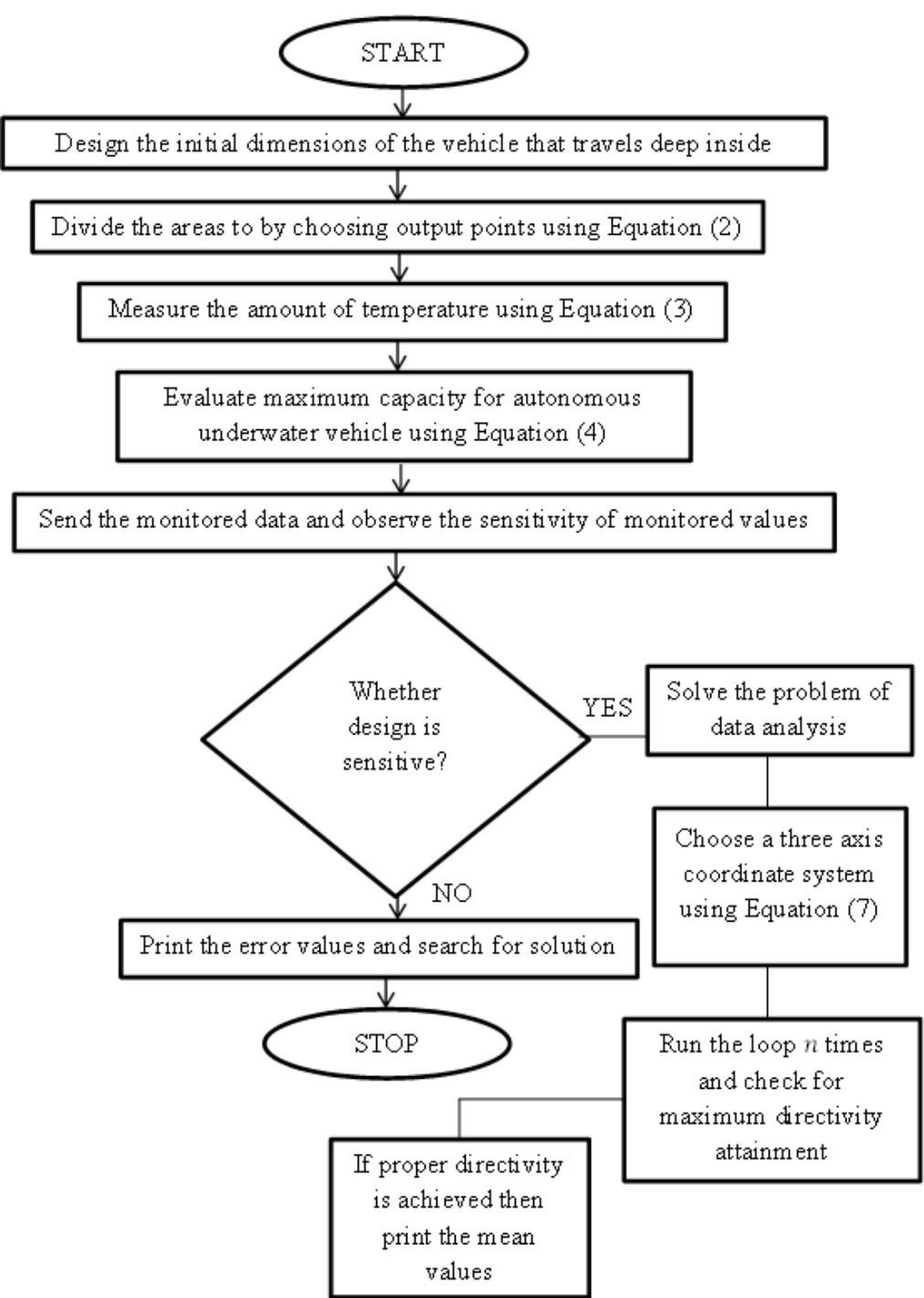

**Figure 2.** Integration of underwater vehicle using hydrophone.

## 4. Results and Discussions

Under two modes of operation, especially offline and online analyses, the proposed implementation on hydrophone for locating proper navigation systems with deep ocean monitoring implications are seen. The findings of the offline analysis are seen using a matching setup in which a boat-shaped vehicle is built, and the observed values are shown using simulated exploration. Furthermore, the findings are presented using a probabilistic path that is determined in an arbitrary manner and covers a maximum depth of 30 metres within the water. Furthermore, the outcomes are examined individually for four different scenarios, providing a significant benefit in developing highly effective solutions. The following are the four different scenario evaluations:

Scenario 1: Discerning output points at different locations;
Scenario 2: Rate of decomposition with respect to unit position;
Scenario 3: Level of sensitivity;
Scenario 4: Motion of directivity;
Scenario 5: Implementation cost.

### Scenario 1

The output points are placed at various places in this scenario, and the end margins of the locations are measured. Hydrophones are thereby incorporated in a moving vehicle within the necessary margins, with output points fixed based on the rate of energy delivered in kilo Watts per hour (kWh). In this situation, enough energy should be delivered at the start of the process to get the vehicle setup to go up to the required place. It is impossible to drive the vehicle to the end points if the energy supplied at the start is insufficient. As a result, even if the energy is underutilised in the beginning, it can be compensated at the conclusion. Thus, in the suggested technique, the beginning energy supply is increased from 40 to 60 kWh, and the final energy supply is reduced using the probabilistic path method since the shortest path to the output sites has been discovered. Figure 3 shows the output points that are measured.

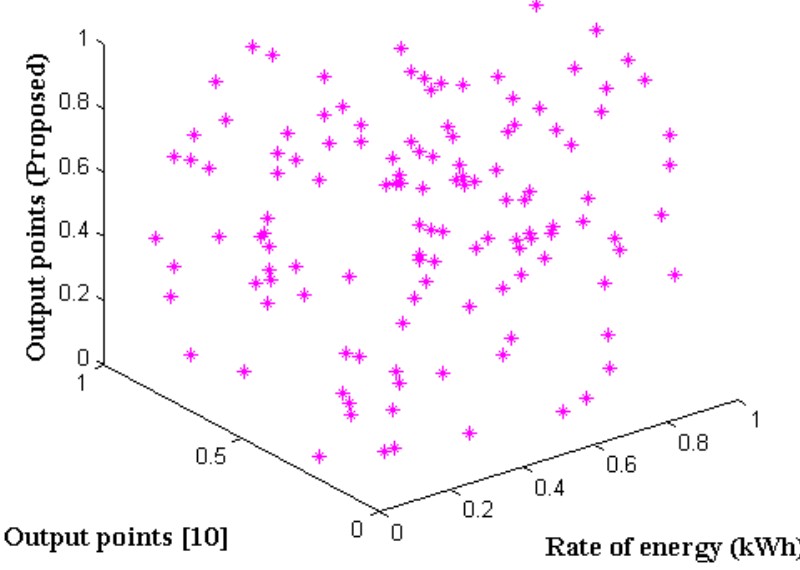

**Figure 3.** Location of output points.

It can be seen in Figure 3 and Table 2 that as the energy rate changes, the output points rise in order to cover the complete distance. Using a probabilistic path finding algorithm, the same strategy has been applied with different formulations that are present in the existing method [10], and only a gradual increase in location expansion has been identified. However, when the proposed formulations are combined with a probabilistic path finding algorithm, it is demonstrated that distance points are correctly located, and that when a high energy rate is applied, the points are converted to a static mode, indicating that the maximum located distance has been reached.

**Table 2.** Output points with energy deviance.

| Rate of Energy (kWh) | Output Points [10] | Output Points (Proposed) |
|:---:|:---:|:---:|
| 60 | 0.8 | 1.7 |
| 120 | 0.5 | 2.8 |
| 180 | 0.3 | 4.8 |
| 240 | 0.9 | 4.8 |
| 300 | 1.4 | 4.8 |

*Scenario 2*

The capacity rate of solar cells should be calculated because solar panels are integrated for providing energy to the hydrophone system. Equation (4) is used to compute the rate of capacity, and the associated decomposition rate is minimised in the suggested method. As a result, 50 different units are used to observe the results, and the decomposition rate is measured in milli-ampere per hour (mAh). Because the speed of supplanting stations in hydrophone is slower, only ten different units are installed at first. However, if the movement grows more rapid, more units are required, and cost per unit cells may be estimated. The main goal of calculating the cost per unit cell is to reduce the number of units used unnecessarily, even if the speed of movement is slowed. As a result, because this scenario is based on dynamic movements, the outputs are observed over time and plotted in Figure 4.

The correct movement of the vehicle setup has been achieved at various time periods, as shown in Figure 4 and Table 3. Despite the fact that vehicle movement is appropriate, the existing method [10] provides a high decomposition rate, indicating that more energy is untapped. However, the proposed method's decomposition rate is lower, and it exists within 150 mAh; with more units installed, a constant decomposition rate can be achieved, and in later stages, a zero decomposition rate can be achieved. In addition, the predicted strategy minimises the time it takes to install units, resulting in lower installation costs.

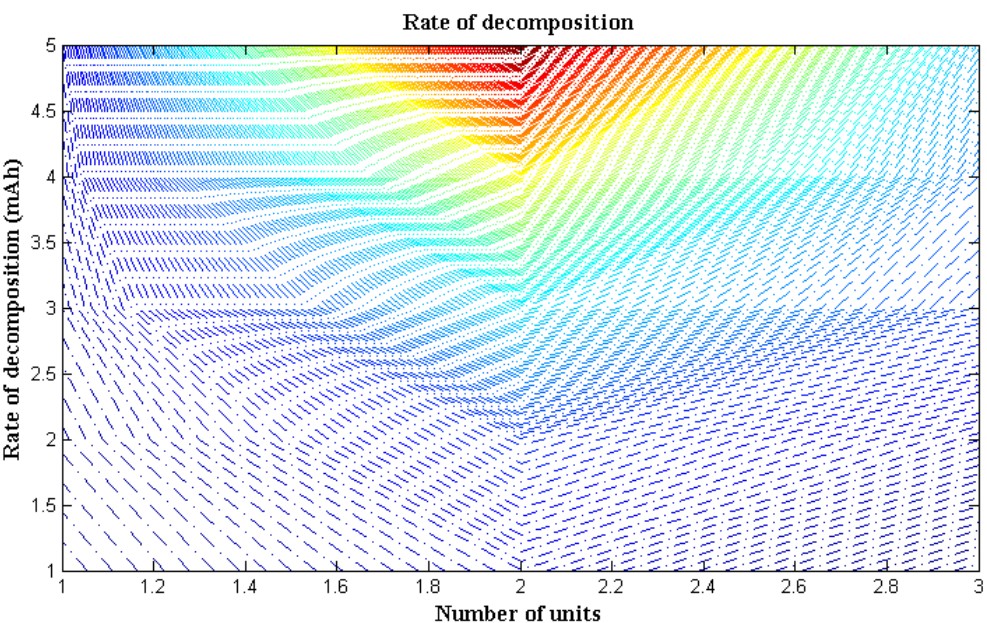

**Figure 4.** Rate of decomposition. The color changes in this figure represents various changes in decomposition rate that varies with number of units. For smaller number of units blue color is represented whereas for increasing units green and red color representations are used.

**Table 3.** Unit decomposition.

| Number of Units | Decomposition Rate mAh [10] | Decomposition Rate mAh [Proposed] |
|---|---|---|
| 10 | 50 | 10 |
| 20 | 83 | 46 |
| 30 | 164 | 89 |
| 40 | 286 | 112 |
| 50 | 458 | 147 |

*Scenario 3*

The sensitivity parameter will be measured after the maximum capacity rate has been observed, as any increase in the rate of decomposition will cause the process to fail during the data transmission phase. As a result, distinct voltage levels are retained for different points, and the hydrophone's dependability is tested and indicated in decibels at each segment (dB). The voltage level for measurement purposes is considered to be between 2 and 12 volts, and this voltage is supplied via internal cell mode with charging. In addition, the reliability rate of installed arrangements must be less than 2 dB, and if it exceeds the corresponding decibel points, the setup is unsuitable for monitoring underground water system activities. In Figure 5, the measured dependability rates of the traditional and new methods are displayed and compared.

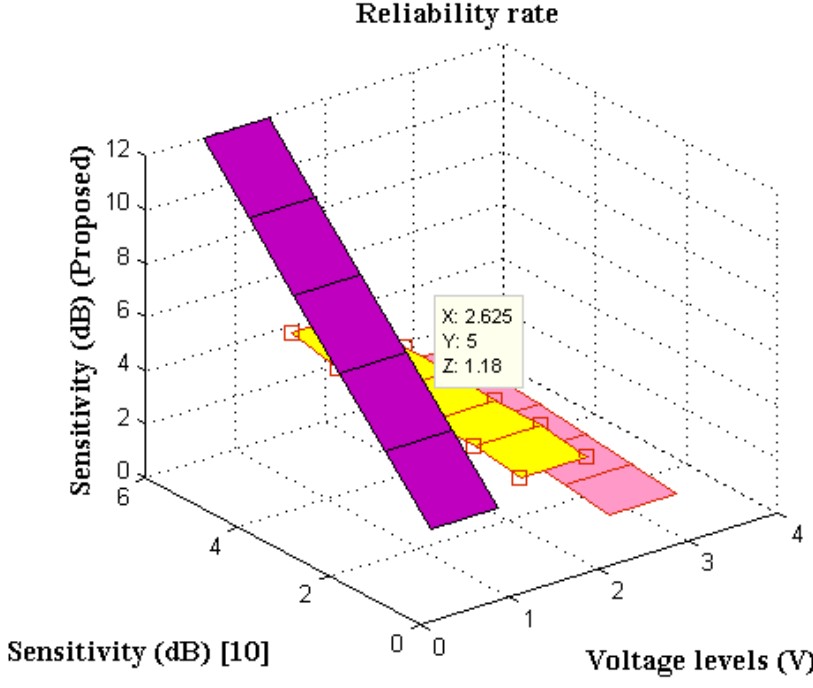

**Figure 5.** Reliability of proposed method.

As can be seen from the comparisons in Figure 5 and Table 4, the existing method is only ascetically suitable for low voltage levels. Even if the voltage level is larger when more installation units are used, the proposed method is more reliable since the reliable points are kept under 1.3 dB.

Equation (6) is used to calculate these reliable points, which take into account all three axis co-ordinates and calculate reliability at exact midpoints. However, if the amount of sensitivity is increased in the operation of monitoring underwater systems, aquatic seeks will be unable to survive, and as a result, the movement of different seeks will be reduced. As a result, the dependability points should be achieved at low voltage levels with commensurate voltage changes.

**Table 4.** Level of detection (sensitivity).

| Voltage Level | Sensitivity (dB) [10] | Sensitivity (dB) [Proposed] |
|---|---|---|
| 2 | 2.86 | 0.47 |
| 4 | 3.14 | 0.64 |
| 6 | 3.19 | 0.86 |
| 8 | 3.27 | 1.03 |
| 10 | 3.32 | 1.18 |
| 12 | 3.69 | 1.29 |

*Scenario 4*

The proposed design is used to check the radiation pattern of a hydrophone antenna in three dimensions. The main goal of determining the radiation pattern is to determine the density of power that should be delivered to the intended model, resulting in static changes according to the incidence angle. Thus, five different angles of incidences ranging from 0 to 360 degrees are considered, with a directivity level of less than one. Furthermore, even though they are at different locations, the directivity of both the transmitting and receiving antennas is checked separately. Signal connection is seen as a wireless medium of communication in which any collisions between connectivity ranges are avoided.

The directivity range of proposed and existing approaches for different incident angles is shown in Figure 6 and Table 5. According to the simulation results, the suggested technique has a lower directivity than the previous method [10], which has a directivity of zero at two angles of incidence ranges. However, using the identical equation, the old technique delivers 2 dB higher directivity. This demonstrates that the proposed method works for all incident angles and frequency rates.

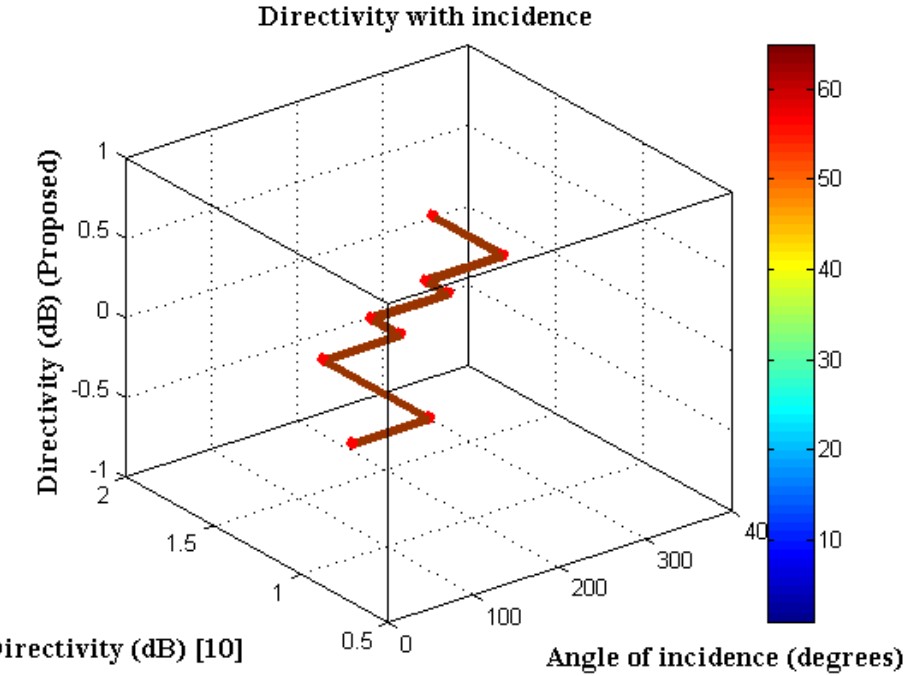

**Figure 6.** Variation of directivity with incident angles.

**Table 5.** Incident directivity.

| Angle of Incidence (Degrees) | Directivity (dB) [10] | Directivity (dB) [Proposed] |
|---|---|---|
| 0 | 0.7 | 0.19 |
| 90 | 1.3 | 0.11 |
| 180 | 1.47 | 0.1 |
| 270 | 1.6 | 0 |
| 360 | 2 | 0 |

*Scenario 5*

In this scenario, the cost of producing a single prototype utilising hydrophone sensor is experimental since it is vital to demonstrate that sophisticated materials are even available at low cost so that everyone in society may buy and assemble it. Even after the assembly process, the same prototype can be turned into a business model and sold to businesses. As a result, the hydrophone sensing units are split and installed independently into eight different categories in order to calculate the implementation cost. The proposed model uses an eight-system concept, which limits the number of sensing units on the bottom of underwater vehicles to 40. More than 40 units are not required in the implementation process because a probabilistic technique is used, and the data units are not covered for cost calculation since the design data units of hydrophone units are covered in the data segment of hydrophone sensors. The cost of underwater vehicles is calculated from the expected values that are represented in Equation (9). The following parameters, such as battery of vehicles, are also included in the calculations where the cost varies with the number of hydrophones that are used for detecting sound waves under the water. Further, the reduction in cost system for underwater vehicles is also based on the surface replication, where for an inflexible surface the vehicle will be damaged. In the proposed method, a secondary surface has been made and the cost of production is also included in such cases. Since a machine learning algorithm is implemented, the fitness function will determine the cost of installed vehicles, but to have probable values Equation (9) is designed. However, the machine learning algorithm does not control the development time of the systems; therefore, only the visibility time periods are added. The major purpose of adding visibility periods is that a high-end camera system must be included for underwater systems as the depth of the ocean is much higher. The implementation cost, which is calculated in roubles, is represented in Table 6.

Figure 7 shows that the total cost of 40 units is calculated in eight rounds, with the suggested technique having a lower initial cost with fewer units than the usual model of underwater monitoring systems [10]. Even with the increasing number of hydrophone sensor units, the cost of underwater hydrophone sensors is substantially lower, and is estimated to be around 25,000 roubles. The previous technology, with the same number of unit divisions, provides a high cost for sensing units of 58,000 roubles.

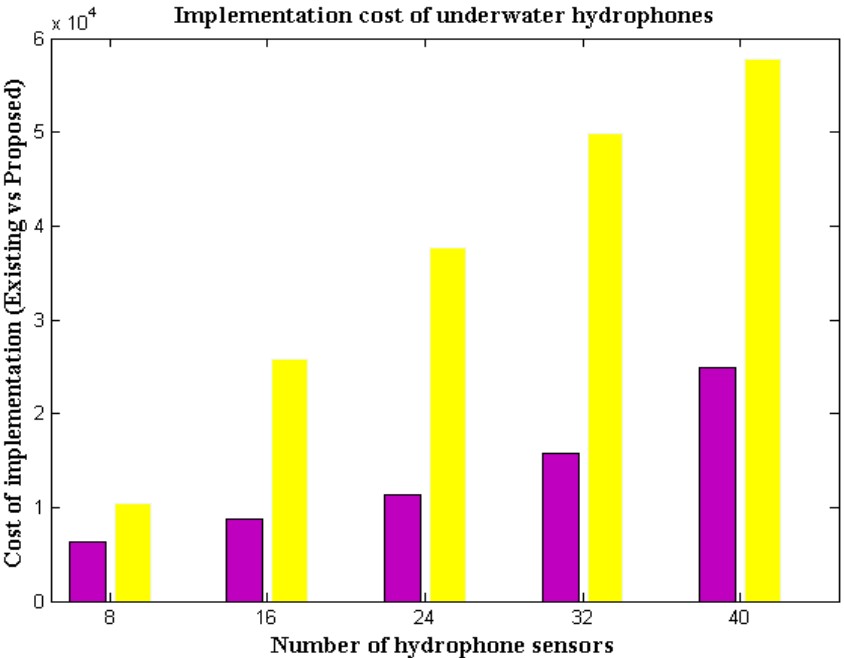

**Figure 7.** Prototype implementation cost.

**Table 6.** Implementation cost.

| Number of Hydrophone Sensors | Cost of Prototype [10] | Cost of Prototype (Proposed) |
| :---: | :---: | :---: |
| 8 | 10,470 | 6320 |
| 16 | 25,780 | 8710 |
| 24 | 37,654 | 11,279 |
| 32 | 49,820 | 15,738 |
| 40 | 57,832 | 24,892 |

### 4.1. Performance Analysis of Probabilistic Determinations

To examine the effectiveness of probabilistic algorithms, a performance analysis has been simulated under real-time conditions with the following two parameters, namely:

- Convergence;
- Robustness.

#### 4.1.1. Convergence Features

For overcoming two independent challenges such as space and time, convergence characteristics monitor the CPU time and memory usage of data units. A multiuser execution system with factor programming steps is chosen because the hydrophone sensors are programmed for external control.

Accurate processing of probabilistic algorithms is thought to require the best convergence characteristics within 80 iterations. The evaluation convergence characteristics of the anticipated and existing methods are shown in Figure 8 [10]. Figure 8 and Table 7 show that the number of iterations is varied between 10 and 100, with the time period of accurate consequence values reported for each iteration. The probabilistic approach that solves a collection of non-linear constructed models converges early within 60 iterations, whereas the previous method can converge at values larger than 80 iterations for the same non-linear model. This demonstrates that, as compared to traditional methods, all complexities in building underwater hydrophone sensors are eliminated.

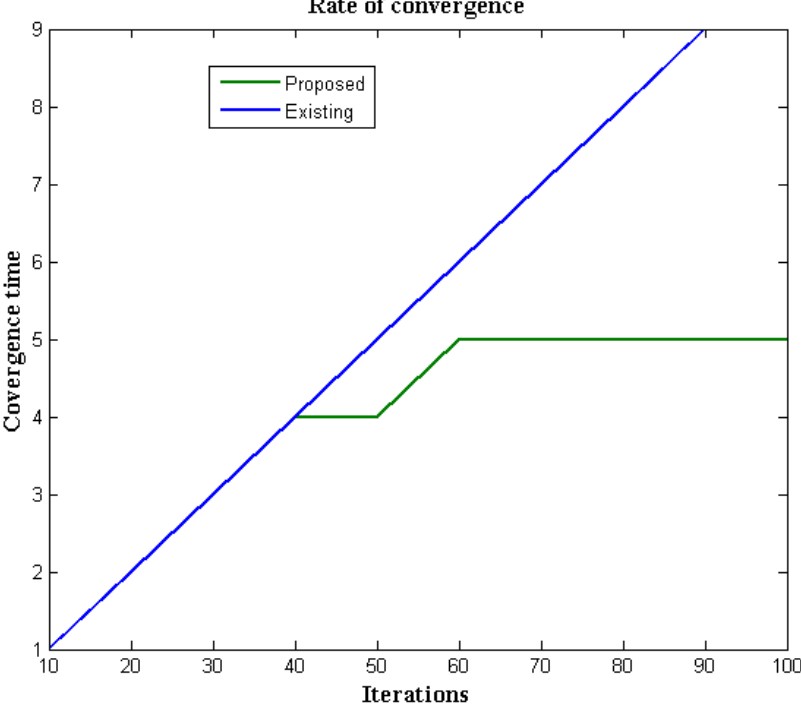

**Figure 8.** Convergence characteristics.

**Table 7.** Convergence characteristics.

| Iteration | Convergence [10] | Convergence (Proposed) |
|---|---|---|
| 10 | 1 | 1 |
| 20 | 2 | 2 |
| 30 | 3 | 3 |
| 40 | 4 | 4 |
| 50 | 5 | 4 |
| 60 | 6 | 5 |
| 70 | 7 | 5 |
| 80 | 8 | 5 |
| 90 | 9 | 5 |
| 100 | 9 | 5 |

### 4.1.2. Control Individualities

The resilience properties of included algorithms can be used to determine the point of global convergence. The technique is best suited for control operations if the global point converges to a local minimal point. As a result, the control design of a probabilistic algorithm is examined and simulated for 10 iteration periods.

An average point is determined in this analysis, and it indicates the fully achievable control actions. Furthermore, if the values are increased at a single focal point, it is logical that data processing units are not controlled; however, for a small number of accomplished values, controllable measures are achieved. Figure 9 and Table 8 show the robustness properties of probabilistic determinations with iterations ranging from one to ten. The average points are achieved in both the existing [10] and proposed methods between the 5th and 10th iteration periods, but the proposed points are much lower than the existing points, indicating that appropriate control measures are achieved in probabilistic algorithms as opposed to other algorithmic determinations.

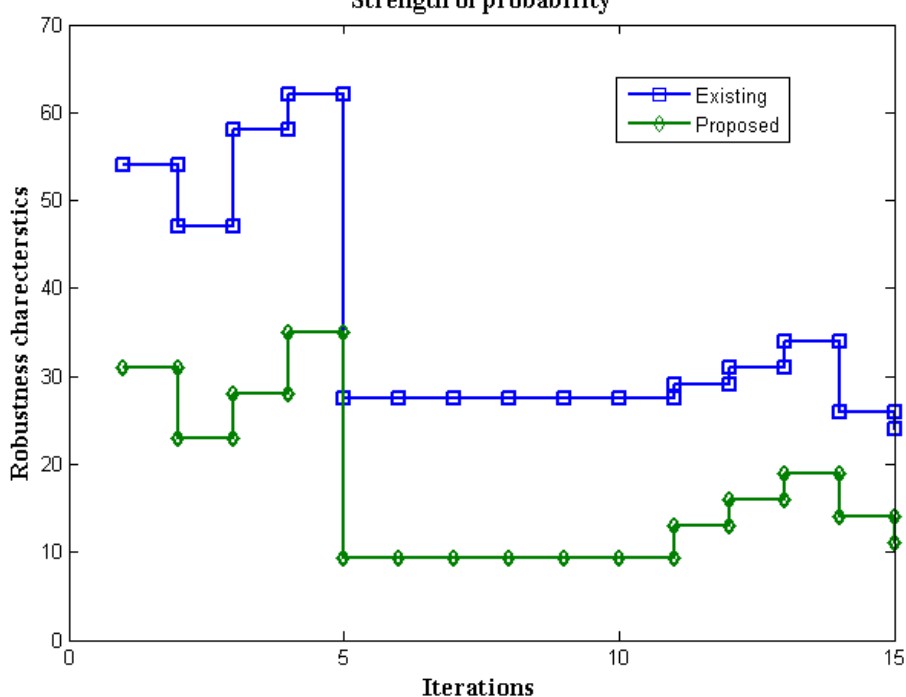

**Figure 9.** Robustness characteristics.

**Table 8.** Strength of probabilistic algorithm.

| Iterations | Robustness [10] | Robustness (Proposed) |
|:---:|:---:|:---:|
| 1 | 54 | 31 |
| 2 | 47 | 23 |
| 3 | 58 | 28 |
| 4 | 62 | 35 |
| 5 | 27.5 | 9.4 |
| 6 | 27.5 | 9.4 |
| 7 | 27.5 | 9.4 |
| 8 | 27.5 | 9.4 |
| 9 | 27.5 | 9.4 |
| 10 | 27.5 | 9.4 |
| 11 | 29 | 13 |
| 12 | 31 | 16 |
| 13 | 34 | 19 |
| 14 | 26 | 14 |
| 15 | 24 | 11 |

### 4.1.3. Computational Complexity

Since probabilistic determinations are made, it is necessary to analyse the time complexities that are present in the underwater systems. Therefore, the number of supplied resources is observed, and only upper bound is observed for solving the overlapping functionalities. In addition, the computational complexities will vary depending on the input functions; thus, only pre-defined inputs are considered in the proposed method. Additionally, three types of complexities which are termed as good, average, and worst scale systems are observed and plotted in Figure 10. From Figure 10, the number of resources and their corresponding complexities are measured, where, as the number of resources is increased, the computational time period is also maximised. Further, the worst-case conditions are observed for all resources, and using the proposed formulations the time complexity is reduced. This can be seen when the number of resources is equal to 20 where at the initial stage the worst complexity is observed as 12.5 s, whereas after connecting the output points, the best time value is achieved which is equal to 6.5 s. This proves that probability distributions in underwater vehicles reduce the time complexity even at high allocated resources.

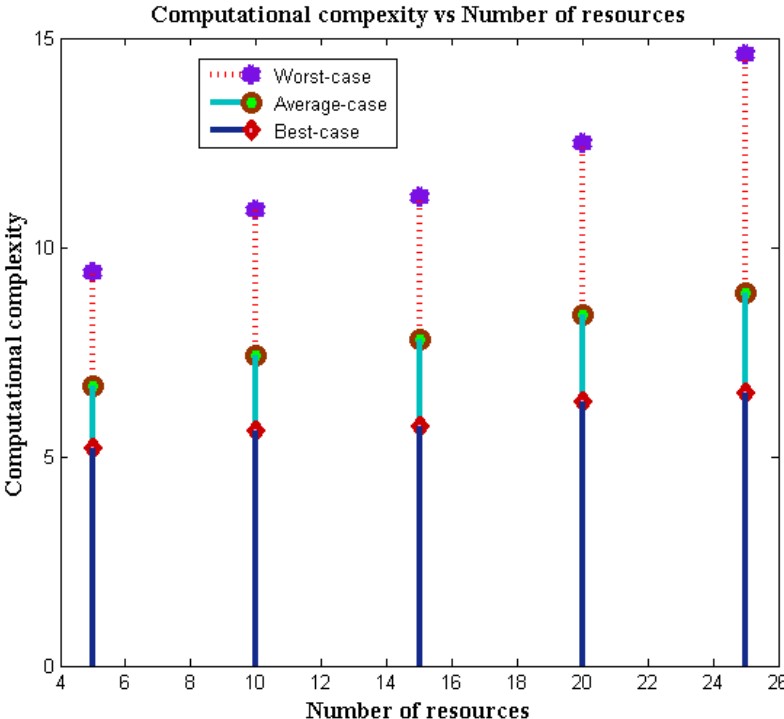

**Figure 10.** Representation of computational complexity.

## 5. Conclusions

To address the various issues that exist in the underground oceanic system, it is necessary to develop a floating boat system that will monitor all underground parameters. As a result, the goal of this research is to create a prototype design that includes a hydrophone as one of the sensing units. If these types of sensing units are included, the cost of constructing underground vehicles will be reduced, allowing all members of society to use them. Furthermore, where no data monitoring system is present, these advanced monitoring systems can be employed to replace conventional cruisers. Furthermore, the path of projection in the resulting design is picked using a unique process known as the probabilistic path finding algorithm. If this algorithm is not integrated, it will be difficult to determine the output units, and the monitoring system's directivity will be incorrect. In comparison to traditional methods, the proposed underwater monitoring system with hydrophone methodology produces satisfactory results in terms of reliability and directivity.

The major findings of the proposed work depend on the upgradation of oceanic systems, as it is not preserved in a proper way. All the systems will follow GPS latitude and longitude points without any measurement monitoring systems. However, if the underwater vehicles are developed with output point units, then deep ocean can be monitored, thus determining exceptional species in the system locations. Further, the integration of the machine learning algorithm reduces the monitoring periods as a boat-shaped vehicle moves at greater speeds. Even the conventional systems are built with plates where the weight will be higher, thus increasing the installation cost. However, the proposed technique has been designed with a structure that reduces the weight of all units, thus making the system to travel further deep inside the oceanic systems. In the future, the underwater vehicles will be designed with reliable sensors where the data gathering possibility will be much higher, and the depth of monitoring will be increased with the reduction in invention weight.

**Author Contributions:** Data curation: A.M.A.; Writing original draft: H.M.; Supervision: P.R.K.; Project administration: S.S. and P.R.K.; Conceptualization: H.M. and P.R.K.; Methodology: S.S., D.S., and H.-N.L.; Validation: A.M.A. and S.S.; Visualization: H.M., D.S. and P.R.K.; Resources: A.M.A. and H.-N.L.; Review and editing: H.M. and P.R.K.; Funding acquisition: S.S., A.M.A. and H.-N.L. All authors have read and agreed to the published version of the manuscript.

**Funding:** This work was supported in part by the National Research Foundation of Korea (NRF) Grant funded by the Korean government (MSIP) (NRF-2021R1A2B5B03002118). This research was supported by the Ministry of Science and ICT (MSIT), Korea, under the ITRC (Information Technology Research Center) support program (IITP-2021-0-01835) supervised by the IITP (Institute of Information & Communications Technology Planning & Evaluation).

**Institutional Review Board Statement:** Not applicable.

**Informed Consent Statement:** Not applicable.

**Data Availability Statement:** Not applicable.

**Conflicts of Interest:** The authors declare no conflict of interest.

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
