# Peer review of "Probabilistic Framework Allocation on Underwater Vehicular Systems Using Hydrophone Sensor Networks"

_water, doi:10.3390/w14081292_

Round 1
Reviewer 1 Report
See the attached review file.

Author Response
Dear learned reviewer, file is attached.

Reviewer 2 Report
I'm sorry, but the English language version of this paper is impossible to understand. There are very many inappropriate words and terms. In the abstract, the word "underground" seems to mean "underwater". The term "hydrophone" is used frequently, but the applications cannot be understood.
If you wish to publish this paper, please find someone who can understand technical English to help with the writing.
Author Response
Q1. I'm sorry, but the English language version of this paper is impossible to understand. There are very many inappropriate words and terms. In the abstract, the word "underground" seems to mean "underwater". The term "hydrophone" is used frequently, but the applications cannot be understood.
If you wish to publish this paper, please find someone who can understand technical English to help with the writing.
Ans: Technical English is corrected in revised version of the Manuscript.
Abstract
This article emphasis the importance on constructing underwater vehicle monitoring system to solve various issues that are related to deep subversive oceans. For solving the issues conventional methods are not implemented whereas a new underwater system is introduced which acts as a sensing device and monitors the rigorous sound of deep ocean systems. However fundamentals of creating underwater vehicles have been considered form conventional systems and the new formulations are generated. This innovative sensing device will function based on the energy produced by the solar cells where low parametric units are installed. Also the energy consumed for operating a particular unit is much lesser and as this results in achieving high reliability using probabilistic path finding algorithm. Further two different application segments have been solved using the proposed formulations including depth of monitoring the ocean. To validate the efficiency of proposed method comparisons have been made with existing methods in terms of navigation output units, rate of decomposition for solar cells, reliability rate and directivity where the proposed method proves to be more efficient for an average percentile of 64%.
Reviewer 3 Report
- The Abstract and Introduction should be revised. Materials could be re-organized for better explanation (see details in reviews).
- Using an algorithm and related computational complexity analysis is expected to be added.
- Background portion needs details by including a more detailed comparisons table. some nice related works can be included like 'A new stable election-based routing algorithm to preserve aliveness and energy in fog-supported wireless sensor networks' or P-SEP algorithm.
- The experiments could be explained in more detail.
- The conclusion could specify the research status and future work clearly.
Author Response

(The authors gave the same response as above.)

Reviewer 4 Report
The work indicates that underwater vehicles are have been adopted for use in oceanic systems in a better manner. The quality of work is good and proposed mathematical model provides in depth insights about different parameters in Underwater vehicles like monitoring the safety precautions with big data and also provides tranquil procedures after safety maintenance. However some important measures needs to be addressed by the authors like what is the Research gap? If the following questions are answered then the paper will be much clear and understandable.
Comments:
- The quality of paper which describes the methodology is good. But the authors need to follow corresponding model for a Research article which indicates that in article Presentation can be improved.
- Standard Research articles should not have any typo errors. The authors should check entire article as there are many small typos in the article. For example in Abstract Comparisons with existing methods in terms of navigation output units, rate of decomposition for solar cells, reliability rate, and directivity were done to validate the efficiency of the suggested method, with the proposed way proving to be more efficient with an average percentile of 64 percent.
- Even though the work is good and considerable. How the work will be justified without providing any information about recent works therefore, the First sentence of the literature work should be cited by some new article. The authors should follow any standard Journal and relevant works should be added.
- What are the fundamentals of implanting underwater vehicles? There is no information regarding the flow in Section 1. The authors should add explanation of basic concepts in 3-4 lines. Unswervingly the research paper does not focus on any fundamentals.
- For implementing the proposed indication whether any big data is captured? Is this the gap identified? The flow of work is good. Since the comparison in Table 1 is very lesser the Authors must add 3-4 relevant papers in comparison table.
- A new mathematical model has been used and it is appreciable. Since the authors have used the mathematical formulations it is necessary to provide equation numbers in correct order. In the paper the format of the equation numbers is not correct. What does the Equations specify?
- In Section 2 the Equations are derived in a direct form. Whether the Equations are considered with basic representations from other papers? If so the Equations should be cited. If not the Equations should be derived. This will provide a good understanding to the readers.
- The following reference can be included in the literature:Underwater Fish Detection and Counting using Improved Mask Region
Author Response
Dear learned reviewer, please find the attached response file.

Round 2
Reviewer 2 Report
Although the revised version of the paper is somewhat improved, there are still major problems with the English expression that make the content impossible to understand. This applies to the abstract and all other sections.
For example, in the abstract, the first sentence ends with "various issues that are related to deep subversive oceans". The word "subversive" means "intending to overthrow, destroy, or undermine an established or existing system, especially a legally constituted government or a set of beliefs." Clearly, this is wrong. What is actually intended?
The second sentence has "which acts as a sensing device and monitors the rigorous sound of deep ocean systems". Why is the sensor monitoring the sound of deep ocean systems? Is it supposed to measure ambient noise? Is it measuring something else?
The next sentence states that the device will have solar cells. Is it supposed to operate on the surface of the ocean, or underwater? There are several vessels that have solar cells to operate on the surface of the ocean and some are used for ocean sensing. There are also some underwater vehicles with solar cells that can operate for short periods underwater. What is intended here?
Similar comments can be made about the text. There are many general statements, but the connection between them is not clear.
Finally, the references do not reveal the intention of this work. They appear to be random. They relate to many different types of research with no common theme.
Decades of work have been done on ocean sensing. The authors should read this work, if it is available to them, before writing a paper of this type. If relevant journals are not available, it may be easier to consider a different field of research.
Author Response
Response file is attached

Reviewer 3 Report
The work is enhanced appealingly. It can be published as is.
Author Response
The authors thank the reviewer for their valuable suggestions